# Association between Menopause, Postmenopausal Hormone Therapy and Metabolic Syndrome

**DOI:** 10.3390/jcm12134435

**Published:** 2023-06-30

**Authors:** Ying-Ju Ou, Jia-In Lee, Shu-Pin Huang, Szu-Chia Chen, Jiun-Hung Geng, Chia-Hung Su

**Affiliations:** 1Department of Pharmacy, Kaohsiung Medical University Hospital, Kaohsiung 80756, Taiwan; ouolive@gmail.com; 2Department of Pharmacology, Graduate Institute of Medicine, College of Medicine, Kaohsiung Medical University, Kaohsiung 80756, Taiwan; 3Department of Psychiatry, Kaohsiung Medical University Hospital, Kaohsiung 80756, Taiwan; u9400039@gmail.com; 4Department of Urology, Kaohsiung Medical University Hospital, Kaohsiung Medical University, Kaohsiung 80756, Taiwan; shpihu73@gmail.com; 5Graduate Institute of Clinical Medicine, College of Medicine, Kaohsiung Medical University, Kaohsiung 80756, Taiwan; 6Department of Internal Medicine, Kaohsiung Municipal Siaogang Hospital, Kaohsiung Medical University, Kaohsiung 81267, Taiwan; scarchenone@yahoo.com.tw; 7Department of Internal Medicine, Division of Nephrology, Kaohsiung Medical University Hospital, Kaohsiung Medical University, Kaohsiung 80756, Taiwan; 8Faculty of Medicine, College of Medicine, Kaohsiung Medical University, Kaohsiung 80756, Taiwan; 9Research Center for Environmental Medicine, Kaohsiung Medical University, Kaohsiung 80756, Taiwan; 10Department of Urology, Kaohsiung Municipal Siaogang Hospital, Kaohsiung Medical University, Kaohsiung 81267, Taiwan; 11Department of Surgery, Kaohsiung Municipal Siaogang Hospital, Kaohsiung 81267, Taiwan

**Keywords:** epidemiologic study, cross-sectional study, metabolic syndrome, menopause, postmenopausal hormone therapy, risk factors

## Abstract

(1) Background: We aimed to explore the associations between menopause, postmenopausal hormone therapy, and metabolic syndrome in a large community-based group of Asian women. (2) Methods: This is a cross-sectional study in which we enrolled women aged 30 to 70 years with sufficient information about menopausal status from the Taiwan Biobank. The definition for metabolic syndrome used in this study aligns with the Bureau of Health Promotion’s (Taiwan) proposed definition. (3) Results: A total of 17,460 women were recruited. The postmenopausal group had a higher metabolic syndrome prevalence (30% vs. 14%) and 1.17 times higher odds ratio (OR) than the premenopausal group (95% confidence interval [CI] = 1.02 to 1.33). Regarding the types of menopause, surgical menopause was associated with metabolic syndrome (OR = 1.40; 95% CI = 1.20 to 1.63); however, natural menopause was not associated with metabolic syndrome. Interestingly, postmenopausal hormone therapy was associated with a lower risk of metabolic syndrome in the women with natural menopause (OR = 0.79; 95% CI = 0.70 to 0.89), but not in those with surgical menopause. (4) Conclusions: Our results suggest that menopause is associated with an increased prevalence of metabolic syndrome, while postmenopausal hormone therapy is associated with a lower prevalence of metabolic syndrome in women with natural menopause.

## 1. Introduction

Metabolic syndrome is a global health problem with an increasing incidence [1]. Approximately 25% of the world’s population is affected by metabolic syndrome [2], and it is estimated that almost 20–30% of middle-aged individuals have the condition [3]. Prevalence rates ranging from 7% to 46% have been reported in women [4,5]. In Taiwan, the estimated prevalence of metabolic syndrome ranges from 10.6% to 30.0% in men and 8.1% to 22.9% in women [6,7].

Previous studies have reported a 60% increased risk of metabolic syndrome in postmenopausal women and an increased incidence of metabolic syndrome during the postmenopausal period [8]. In addition, individuals diagnosed with metabolic syndrome are at a significantly higher risk of experiencing fatal heart attacks or strokes, as well as developing type 2 diabetes mellitus (DM), with nearly twice the likelihood of the former and five times the likelihood of the latter compared to those without metabolic syndrome [2].

Metabolic syndrome is characterized by a combination of metabolic dysfunctions, such as glucose intolerance, low levels of high density lipoprotein cholesterol (HDL-C), high levels of triglycerides (TGs), obesity, and hypertension [9]. In addition, metabolic syndrome is linked to a heightened risk of cardiovascular disease (CVD) [10,11], attributable to an array of contributory risk factors such as elevated glucose levels, hypertension, dyslipidemia, insulin resistance, and obesity (particularly central obesity) [12]. As per the National Cholesterol Education Program Adult Treatment Panel III (NCEP ATPIII) criteria, the diagnosis of metabolic syndrome may be established by the presence of at least three of the aforementioned components [13].

Metabolic syndrome and its constituent components, including hypertension, exhibit a higher prevalence in older women compared to men, potentially due to factors such as menopausal transition, ovarian senescence, hormonal fluctuations, and increased body mass index [14]. The majority of metabolic syndrome components are adversely impacted by menopause, with surgical menopause being potentially associated with a greater incidence of metabolic syndrome in comparison to natural menopause [15]. Menopausal hormone therapy has been shown to be the most effective treatment for women suffering from menopausal symptoms [16,17], and estrogen deficiency has been shown to be a key factor in the development of metabolic syndrome in postmenopausal women [18,19].

Metabolic syndrome is an important issue because it increases the risk of CVD, DM, and mortality. Menopause has been associated with metabolic syndrome, and postmenopausal hormone therapy appears to reduce the prevalence of metabolic syndrome. Research investigating variances in the prevalence and characteristics of metabolic syndrome according to menopausal status is crucial, but few studies have been conducted in Asian women. Given that community-based statistics are imperative for the effective prevention and management of metabolic syndrome in postmenopausal women, the aim of this study was to investigate the associations between menopause, postmenopausal hormone therapy, and metabolic syndrome in a sizable population of Asian women.

## 2. Materials and Methods

### 2.1. Study Population

This is a cross-sectional study, and data for the present study were obtained from the Taiwan Biobank (TWB), an extensive community-based research database that has been collecting information from cancer-free volunteers aged 30 to 70 years since 2008. The TWB has recruited individuals from 29 centers throughout Taiwan. The detailed profile and methods regarding the development of the TWB have been described in previous studies [20,21,22]. Participants with missing data on menopausal status and confounders (N = 132) were excluded. Finally, a total of 17,460 participants with adequate baseline data were enrolled, and they were classified into two groups based on menopausal status: premenopausal and postmenopausal groups. The detailed enrollment is shown in Figure 1. This study was approved by our institute (KMUHIRB-E(I)-20210058), and written informed consent was obtained from all participants. All investigations were conducted according to the Declaration of Helsinki.

### 2.2. Variables

To gather the variables analyzed, we utilized diverse data sources, including questionnaires, physical examinations, and blood tests. The questionnaires encompassed a wide range of factors, such as age, gender, smoking, drinking, coffee consumption habits, exercise routines, educational status, marital status, and medical history. Physical examinations were conducted to assess waist circumference, body mass index, and blood pressure. Blood tests provided data on metabolic profiles, including fasting glucose, cholesterol-related profiles, and triglycerides. The data were collected from participants at the time of their enrollment in the study, which spanned from December 2008 to February 2020.

### 2.3. Premenopausal, Postmenopausal, Type of Menopause, and Postmenopausal Hormone Therapy Assessments

All of the participants completed a questionnaire on female physiology including questions about menstrual health such as the age at onset of menstruation, menstrual cycle regularity, and current menstrual status. Participants who reported ongoing menstruation were classified as being premenopausal (premenopausal group). The participants who reported no menstruation were further queried as to whether they had menstruated in the 12 months since their last menstrual cycle, and, if not, they were classified as being menopausal (postmenopausal group). The postmenopausal group was then subdivided into two categories: natural menopause and surgical menopause. The questionnaire also surveyed the use of postmenopausal hormone medications. Participants who reported the regular use of these drugs for a duration of more than 6 months were categorized as users, while those who did not meet this criterion were categorized as non-users.

### 2.4. Definition and Assessment of Metabolic Syndrome

Metabolic syndrome was diagnosed as per the definition proposed by the Bureau of Health Promotion (Taiwan) in 2006, based on the manifestation of three or more of the following five characteristics [20]: (1) abdominal obesity (waist circumference ≥ 80 cm, based on an Asian population), hypertriglyceridemia (fasting serum TG level ≥ 150 mg/dL), low HDL-C (fasting serum HDL-C < 50 mg/dL), elevated blood pressure (systolic blood pressure ≥ 130 mm Hg or diastolic blood pressure ≥ 85 mm Hg), and elevated fasting glucose (fasting serum glucose ≥ 100 mg/dL). All of the participants also underwent related physical examinations and blood tests.

### 2.5. Statistical Analyses

Continuous variables, such as age, waist circumference, body mass index, systolic and diastolic blood pressure, fasting glucose, TG, and HDL-C, were described using means and standard deviations (±). Categorical variables, such as smoking, alcohol status, habitual coffee consumption, physical activity, marital status, educational status, gravidity, parity, abortion, breastfeeding, use of postmenopausal hormone medications, and history of chronic kidney disease, were described using the numbers and percentages of the participants. To compare variables between the premenopausal and postmenopausal groups, we used the independent *t*-test for continuous variables and the chi-square test for categorical variables. To investigate the association between metabolic syndrome and postmenopausal status, we used logistic regression models to calculate odds ratios (ORs) and their corresponding confidence intervals (CIs). Given that age is a strong independent risk factor for metabolic syndrome, we controlled for age as a fixed confounder [23]. Other confounding variables were identified based on a literature review, and included smoking [24,25], alcohol consumption [26], habitual coffee consumption [24], physical activity [27], marital status [28], educational status [23], gravidity [29], parity [29], abortion [29], breastfeeding [30], the use of postmenopausal hormone medications [31], and history of chronic kidney disease [32]. We applied similar statistical methods to estimate the OR for metabolic syndrome by menopausal type. We reported *p*-values to indicate the statistical significance of differences observed between the two groups, with a significance level of *p* < 0.05. All statistical analyses were performed using R version 4.2.3 (R Foundation for Statistical Computing, Vienna, Austria) and SPSS version 20.0 (IBM Inc., Armonk, NY, USA).

## 3. Results

### 3.1. Clinical Characteristics of the Study Participants

The present study included 17,460 women, of whom 5976 (34.3%) were premenopausal and 11,484 (65.7%) were postmenopausal. The mean ages were 44 ± 6 and 61 ± 6 years in the premenopausal and postmenopausal groups, respectively (Table 1). The postmenopausal women were significantly older, had a larger waist circumference, higher blood pressure, higher rates of marriage, gravidity, parity, abortion, and a lower prevalence of smoking and alcohol consumption than the premenopausal women (Table 1). In addition, the postmenopausal women had significantly higher fasting glucose and TG levels, but no significant difference in HDL-C level compared to the premenopausal women. Regarding the etiology of menopause, most of the women experienced natural menopause, and a smaller proportion underwent surgical menopause. Regarding the history of chronic kidney disease, the postmenopausal women had a significantly higher prevalence.

### 3.2. Associations between Menopause, Metabolic Syndrome, and Its Components

The overall prevalence of metabolic syndrome in the studied population was 24.7%, including 14% in the premenopausal group and 30% in the postmenopausal group (Table 2). In binary regression analysis adjusting for age, the postmenopausal group was associated with a 1.17-fold increased risk of metabolic syndrome (OR, 1.17; 95% CI, 1.03 to 1.33) (Table 2). After adjusting for potential epidemiological variables including age, sex, smoking status (never vs. ever), alcohol status (never vs. ever), marital status (yes vs. no), and educational status, the risk remained the same (OR, 1.17; 95% CI, 1.02 to 1.33) (Table 2). Similarly, after adjusting for age, the postmenopausal group status was associated with a 1.33-fold increased risk of impaired glucose tolerance (OR, 1.33; 95% CI, 1.15 to 1.53) and a 1.83-fold increased risk of hypertriglyceridemia (OR, 1.83; 95% CI, 1.59 to 2.11) (Table 2). However, there were no significant differences in the prevalence of hypertension, waist circumference, and low HDL-C between the premenopausal and postmenopausal groups (Table 2).

### 3.3. Associations between the Types of Menopause and Metabolic Syndrome

To further compare the prevalence of metabolic syndrome between the premenopausal women and the women with surgical and natural menopause, we used a multivariable regression model. A significant association was observed between the prevalence of metabolic syndrome and surgical menopause (OR, 1.40; 95% CI, 1.20 to 1.63) (Table 3), but not with natural menopause (OR, 1.10; 95% CI, 0.96 to 1.23) (Table 3).

### 3.4. Association between Postmenopausal Hormone Therapy and Metabolic Syndrome in the Postmenopausal Women

We further explored the association between postmenopausal hormone therapy and the prevalence of metabolic syndrome in the women with menopause, as shown in Table 4. The results indicated that users of postmenopausal hormones had a significantly lower prevalence of metabolic syndrome than non-users among the women with natural menopause (age-adjusted OR, 0.78; 95% CI, 0.69 to 0.87; multivariable OR, 0.79; 95% CI, 0.70 to 0.89). However, no significant association was found between postmenopausal hormone therapy and metabolic syndrome among the women with surgical menopause (age-adjusted OR, 0.83; 95% CI, 0.67 to 1.02; multivariable OR, 0.84; 95% CI, 0.68 to 1.05) (Table 4).

## 4. Discussion

Our study provides a population-based analysis of a large group of perimenopausal Asian women (*n* = 17,460). The study findings suggest that menopause is associated with a significantly higher risk of metabolic syndrome, especially for those with surgical menopause. In addition, postmenopausal hormone therapy was found to be associated with a lower prevalence of metabolic syndrome in the women with natural menopause. This study is the largest to date of the associations among menopausal status, postmenopausal hormone therapy, and metabolic syndrome in Asian women. Our findings could be relevant and informative for researchers and healthcare professionals in the field of women’s health.

The prevalence of metabolic syndrome in postmenopausal women greatly varies across different countries, studies, and diagnostic criteria, ranging from 13.0% in India to 69.0% in Iran [33,34,35,36,37,38,39,40,41,42,43,44,45,46,47,48,49,50,51]. In the present study, the prevalence of metabolic syndrome was 30% among postmenopausal women, which is similar to several previous studies, including one conducted in China by Ruan et al. [46]. In their study, the prevalence of metabolic syndrome in postmenopausal women was 33.7% [46], while in a study of 2671 Korean postmenopausal women, 54.6% had metabolic syndrome [38]. In addition, a study in Canada found that 32.6% of postmenopausal women met the criteria for metabolic syndrome [51]. In comparison, studies in the USA (13.7%) [39] and India (13%) [33] have reported the lowest prevalence of metabolic syndrome in postmenopausal women, while studies in Iran (69% and 64%) [34,44] have reported the highest prevalence of metabolic syndrome in postmenopausal women. Differences in age, ethnicity, sociodemographic and genetic factors, lifestyle, type of menopause (natural/surgical), marital status, educational status, breastfeeding, and the criteria used to define metabolic syndrome may explain this variability.

We also found that the postmenopausal women had a 1.17-fold higher risk of metabolic syndrome compared to the premenopausal women, which is consistent with previous studies. A study of 12,611 women conducted in 2020 found that the adjusted relative risk of metabolic syndrome was 1.10 times higher in postmenopausal women compared to premenopausal women (95% CI, 1.01 to 1.19) [51]. Similarly, another study that included 958 women found that the odds of metabolic syndrome were 2.75 times higher in postmenopausal women compared to premenopausal women [50]. In addition, a meta-analysis of 23 studies involving 66,801 women found that postmenopausal women had a 3.54-fold higher risk of metabolic syndrome compared to premenopausal women [52].

Regarding the individual components of metabolic syndrome in relation to menopausal status, our results indicated that the postmenopausal women had a significantly higher prevalence of impaired glucose tolerance and hypertriglyceridemia compared to the premenopausal women, even after adjusting for potential confounding factors. However, no significant differences were found with regards to waist circumference, hypertension, and low HDL-C levels. Impaired glucose tolerance is a crucial component of metabolic syndrome, and previous studies have established a clear association between impaired glucose tolerance and menopause. One study conducted in Taiwan found that the number of years since menopause was an independent risk factor for impaired glucose tolerance (OR, 1.05; 95% CI, 1.01 to 1.08) [53], while another study reported a higher prevalence of impaired glucose tolerance in postmenopausal women, which was partially explained by changes in insulin sensitivity and beta-cell function during menopause [54]. Several studies have also reported that menopause is associated with higher TG levels, independent of other risk factors such as age, body mass index, and smoking status. A meta-analysis of 66 studies with over 114,655 women found that postmenopausal women had higher levels of TGs than premenopausal women (0.27 mmol/L, 95% CI, 0.22 to 0.31) [55]. Even though we did not find significant differences in waist circumference, hypertension, and low HDL-C levels between the premenopausal and postmenopausal women, other studies have reported an association between menopause and these components of metabolic syndrome. A previous meta-analysis found that all components of metabolic syndrome, except for HDL-C, were significantly worse in postmenopausal women [15]. The relationships between menopause and individual components of metabolic syndrome are complex and multifactorial, and further research is needed to clarify the underlying mechanisms and potential interventions for managing metabolic syndrome in postmenopausal women.

An interesting finding of this study is that the type of menopause may impact the prevalence of metabolic syndrome, and the women who underwent surgical menopause (33%) had a higher risk of metabolic syndrome than those who underwent natural menopause (30%) and those in the premenopausal group (14%). These findings are consistent with a previous study that analyzed 1052 women and found a 1.52-fold increased risk of metabolic syndrome in the surgical menopause group compared to the premenopausal group [56]. However, two other studies did not reveal significant differences in the prevalence of metabolic syndrome between surgical menopause and premenopausal groups [57,58]. A possible explanation for the different findings in these studies may be that they had relatively small sample sizes, so their results may not be generalizable to all women. Despite some conflicting results, most studies suggest that there is an increased risk of metabolic syndrome following surgical menopause. Further research is needed to fully understand the relationship between surgical menopause and metabolic syndrome, as well as to identify factors that may influence this association.

Our results also showed that the women who used postmenopausal hormone therapy had a lower risk of developing metabolic syndrome during natural menopause. Although several studies have reported mixed results, overall, the evidence suggests that hormone therapy may have a protective effect against metabolic syndrome in women who undergo natural menopause [31,39,59,60,61]. A 2019 study analyzed data from 39,295 postmenopausal women and found that those who used postmenopausal hormone therapy had a significantly lower risk of metabolic syndrome compared to those who did not use postmenopausal hormone therapy [59]. Other studies have also demonstrated similar findings, and reported that postmenopausal hormone therapy is associated with lower levels of insulin resistance, better lipid profiles, and a lower risk of type 2 DM [31]. However, there is currently no detailed research regarding the use of hormone therapy to prevent the occurrence of metabolic syndrome in postmenopausal women, and so it is currently not recommended to use such therapy in menopausal women to prevent metabolic syndrome [62]. Therefore, individualized medical treatment for postmenopausal women to reduce the factors associated with postmenopausal syndrome, chronic diseases, and metabolic syndrome is the best strategy at present [62].

Another noteworthy finding from our study is that postmenopausal hormone therapy did not reduce metabolic syndrome in women with surgical menopause compared to natural menopause. It is well-established that surgical menopause is characterized by a heightened prevalence of metabolic syndrome, obesity, and lipid abnormalities in contrast to natural menopause [15,63]. The underlying mechanisms for this disparity involve abrupt hormonal changes, a persistent low-grade inflammatory state following surgery, and modifications in adipose tissue distribution and lipid metabolism resulting from the surgical procedure [15]. However, no research has specifically focused on the effects of postmenopausal hormone therapy in this population, and our study was the first to explore this topic. We proposed three reasons to explain these findings. Firstly, the timing of postmenopausal hormone therapy initiation differs between natural and surgical menopause. Natural menopause allows for a shorter gap between hormone decline and postmenopausal hormone therapy initiation, preserving metabolic function [64]. Surgical menopause involves an abrupt loss of ovarian function, potentially leading to a longer gap and a less favorable metabolic response [64]. Secondly, underlying health conditions, such as cancer or severe gynecological disorders, in surgical menopause may increase the risk of metabolic syndrome [65]. Hormone therapy may not fully counteract the metabolic changes associated with underlying health conditions in surgical menopause. Thirdly, hormone levels decline gradually in natural menopause, allowing the body to adapt, while surgical menopause causes an immediate loss of hormones [66]. Further studies are needed to validate these findings and provide a more comprehensive understanding of the effects of postmenopausal hormone therapy in women with surgical menopause.

The exact mechanism by which menopause increases the risk of metabolic syndrome is not fully understood, but it is believed to be related to hormonal changes that occur during menopause [62,67,68]. Estrogen is known to play a role in regulating insulin sensitivity and glucose metabolism. During menopause, estrogen levels decline, which can lead to insulin resistance. This can then result in elevated blood sugar levels, which can contribute to the development of metabolic syndrome [14,69]. In addition, abdominal fat increases during menopause, which is a risk factor for metabolic syndrome. Abdominal fat is metabolically active and produces hormones and inflammatory molecules that can contribute to insulin resistance, high blood pressure, and other components of metabolic syndrome [70]. Other hormonal changes that occur during menopause, such as a decrease in levels of the hormone adiponectin, which is involved in regulating glucose and lipid metabolism, may also play a role in the development of metabolic syndrome [71].

There are some limitations in our study, including its cross-sectional design and the fact that some of the data were derived from a self-reported questionnaire. In addition, some relevant factors associated with metabolic syndrome were missing. The frequency and intensity of physical activity were not reported, and information regarding the timing, methods, and frequency of hormonal therapy use was also not collected. According to the guidelines provided by the National Institute for Health and Care Excellence (NICE) on Menopause Diagnosis and Management, hormone therapy should be prescribed at the lowest effective dose and for the shortest duration necessary to alleviate menopausal symptoms [66]. The specific duration of administration can vary depending on individual needs and the symptoms being addressed. Various methods are available for hormone therapy administration, including oral, transdermal, and vaginal routes [66]. The choice of method depends on factors such as the individual’s symptoms, preferences, and medical history [66]. In cases where a woman has undergone a hysterectomy, estrogen alone can be prescribed. However, for women with an intact uterus, a combination of estrogen and progestogen (progesterone or a progestin) is usually required to protect the lining of the uterus [66]. It is important for future research to consider these details when studying hormone therapy and its effects on the prevention of metabolic syndrome. Selection bias from the database may have occurred as a result of participant recruitment and the applied exclusion criteria. In addition, the psychological factors, such as anxiety and depression, which may alter postmenopausal hypertension were not examined [72]. The cause of metabolic disorders and their associated symptoms, including obesity, is multifaceted and potentially undervalued in terms of ethnicity, with specific populations potentially exhibiting a greater susceptibility to diseases caused by obesity. In addition, plasma renin activity and androgen concentration were not investigated, although androgens can contribute to the activation of the renin–angiotensin system by increasing intrarenal angiotensinogen, and they can also play a role in increasing blood pressure [72]. Finally, the type of delivery of menopausal hormone therapy (oral or transdermal), duration of use, and whether a high or low dose is used may play a role in its efficacy; however, current evidence is not consistent with regards to which delivery mode is more effective.

## 5. Conclusions

Our findings indicate that menopause is associated with an increased risk of metabolic syndrome, underscoring the importance of monitoring, and implementing preventive interventions for menopausal women. Furthermore, our analysis showed that postmenopausal hormone therapy was associated with a lower risk of metabolic syndrome among the women who underwent natural menopause, suggesting a potential protective effect of hormone therapy against metabolic syndrome. Nonetheless, more studies are needed to validate the effectiveness of hormone therapy in reducing the risk of metabolic syndrome, and to determine the optimal dose, duration, and type of hormone therapy.

## Figures and Tables

**Figure 1 jcm-12-04435-f001:**
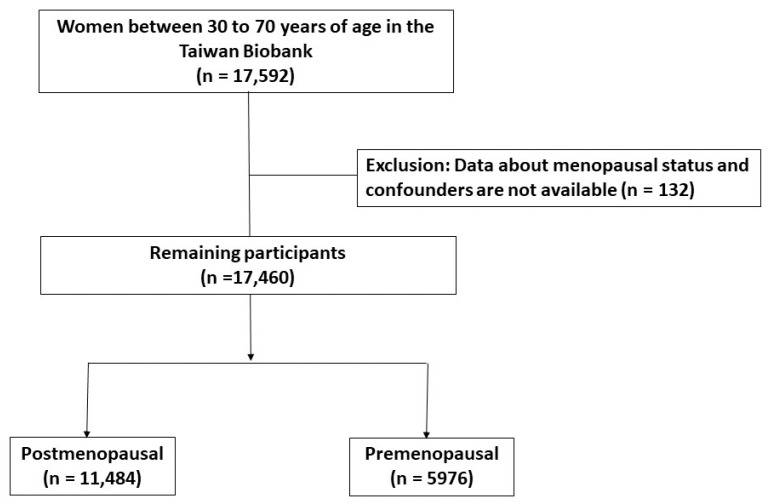
Study participants.

**Table 1 jcm-12-04435-t001:** Clinical profiles of study population (*n* = 17,460).

Characteristics	Premenopausal	Postmenopausal	*p* Value
Number of participants	5976	11,484	-
Age, year	44 ± 6	61 ± 6	<0.001
Waist circumference, cm	80 ± 10	83 ± 10	<0.001
Body mass index, kg/m^2^	23.6 ± 3.9	23.9 ± 3.6	<0.001
Systolic blood pressure, mmHg	113 ± 16	126 ± 20	<0.001
Diastolic blood pressure, mmHg	70 ± 11	73 ± 11	<0.001
Smoke, ever, *n* (%)	675 (11)	609 (5)	<0.001
Alcohol status, ever, *n* (%)	224 (4)	325 (3)	0.001
Habitual coffee consumption, *n* (%)	2830 (47)	4419 (39)	<0.001
Physical activity, yes, *n* (%)	1672 (28)	6664 (58)	0.394
Married, yes, *n* (%)	5100 (85)	10,976 (96)	<0.001
Educational status ≥ college, *n* (%)	3813 (64)	3628 (32)	<0.001
Gravidity, yes, *n* (%)	5006 (84)	10,834 (94)	<0.001
Parity, yes, *n* (%)	4824 (81)	10,680 (93)	<0.001
Abortion, yes, *n* (%)	2977 (50)	7150 (62)	<0.001
Breastfeeding, *n* (%)	3473 (63)	7150 (62)	0.633
Hormone therapy use, *n* (%)	472 (8)	2304 (20)	<0.001
Fasting glucose, mg/dL	91 ± 17	98 ± 21	<0.001
Triglyceride, mg/dL	95 ± 103	118 ± 74	<0.001
High-density lipoprotein cholesterol, mg/dL	58 ± 13	58 ± 13	0.131
Menopause etiology			
Natural		9441 (82)	
Surgical		1967 (17)	
Without information		76 (1)	
History of chronic kidney disease, *n* (%)	13 (0)	258 (2)	<0.001

**Table 2 jcm-12-04435-t002:** Odds ratios for the presence of metabolic syndrome and its components by menopausal status.

	Premenopausal (*n* = 5976)	Postmenopausal (*n* = 11,484)	*p* Value
Presence of metabolic syndrome, *n* (%)	841 (14)	3471 (30)	
Age-adjusted OR (95% CI)	1.0 (reference)	1.17 (1.03 to 1.33)	0.019
Multivariable OR (95% CI)	1.0 (reference)	1.17 (1.02 to 1.33)	0.022
Presence of hypertension, *n* (%)	1048 (18)	5425 (47)	
Age-adjusted OR (95% CI)	1.0 (reference)	0.97 (0.86 to 1.09)	0.562
Multivariable OR (95% CI)	1.0 (reference)	0.96 (0.85 to 1.08)	0.490
Presence of impaired glucose tolerance, *n* (%)	570 (10)	2991 (26)	
Age-adjusted OR (95% CI)	1.0 (reference)	1.33 (1.15 to 1.53)	<0.001
Multivariable OR (95% CI)	1.0 (reference)	1.30 (1.12 to 1.50)	<0.001
Presence of increased waist circumference, *n* (%)	2812 (47)	6908 (60)	
Age-adjusted OR (95% CI)	1.0 (reference)	1.03 (0.93 to 1.15)	0.562
Multivariable OR (95% CI)	1.0 (reference)	1.04 (0.94 to 1.16)	0.438
Presence of hypertriglyceridemia, *n* (%)	696 (12)	2554 (22)	
Age-adjusted OR (95% CI)	1.0 (reference)	1.83 (1.59 to 2.11)	<0.001
Multivariable OR (95% CI)	1.0 (reference)	1.84 (1.60 to 2.12)	<0.001
Presence of low high-density lipoprotein cholesterol, *n* (%)	1607 (27)	3281 (29)	
Age-adjusted OR (95% CI)	1.0 (reference)	1.03 (0.91 to 1.15)	0.669
Multivariable OR (95% CI)	1.0 (reference)	1.04 (0.92 to 1.17)	0.558

OR = odds ratio; CI = confidence interval. Multivariable analysis adjusts for age, smoke status, alcohol status, habitual coffee consumption, physical activity, marital status, educational status, gravidity, parity, abortion, breastfeeding, hormone therapy use, and history of chronic kidney disease.

**Table 3 jcm-12-04435-t003:** Odds ratios for the presence of metabolic syndrome by the types of menopause.

Type of Menopause	Presence of Metabolic Syndrome, *n* (%)	Age-Adjusted OR (95% CI)	Multivariable OR (95% CI)
Premenopause	841 (14)	1.0 (reference)	1.0 (reference)
Natural menopause	2812 (30)	1.11 (0.97 to 1.26)	1.10 (0.96 to 1.23)
Surgical menopause	642 (33)	1.38 (1.19 to 1.60)	1.40 (1.20 to 1.63)

OR = odds ratio; CI = confidence interval. Multivariable analysis adjusts for age, smoke status, alcohol status, habitual coffee consumption, physical activity, marital status, educational status, gravidity, parity, abortion, breastfeeding, hormone therapy use, and history of chronic kidney disease.

**Table 4 jcm-12-04435-t004:** Age and multivariable-adjusted odds ratios for the presence of metabolic syndrome by postmenopausal hormone use in women with menopause.

Postmenopausal Hormone Use	Age-Adjusted OR (95% CI)	*p* Value	Multivariable OR (95% CI)	*p* Value
Women with natural menopause				
Non-users	1.00 (reference)		1.00 (reference)	
Users	0.78 (0.69 to 0.087)	<0.001	0.79 (0.70 to 0.89)	<0.001
Women with surgical menopause				
Non-users	1.00 (reference)		1.00 (reference)	
Users	0.83 (0.67 to 1.02)	0.079	0.84 (0.68 to 1.05)	0.124

OR = odds ratio; CI = confidence interval. Multivariable analysis adjusts for age, smoke status, alcohol status, habitual coffee consumption, physical activity, marital status, educational status, gravidity, parity, abortion, breastfeeding, hormone therapy use, and history of chronic kidney disease.

## Data Availability

The raw data supporting the conclusions of this article will be made available by the authors, without undue reservation.

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
