# Peer review of "Association between Menopause, Postmenopausal Hormone Therapy and Metabolic Syndrome"

_jcm, 2023, doi:10.3390/jcm12134435_

Round 1
Reviewer 1 Report
It is an interesting study that analyzes the association among menopausal status, hormonal therapy, and metabolic syndrome, with a cross-sectional design of data obtained from a cohort. The results bring new knowledge about this association, with the strength of a large sample and limitations that are recognized by the authors.
It is a cross-sectional study of data from a cohort; therefore, I think it needs to be clarified in the study design and not just pointed out as a limitation of the study. In all the text this precision must be made since the term “cohort” implies a longitudinal study and analysis, which was not done.
Also, it is necessary to specify how was obtained the information about the sociodemographic and health status variables in methods section. In the study population, it should be pointed out until the year the data were obtained.
Line 159: If it is a cross-sectional, the word “cohort” generates confusion.
Line 204. The analysis is not a cohort, it is a cross-sectional analysis of a cohort.
Author Response
- It is a cross-sectional study of data from a cohort; therefore, I think it needs to be clarified in the study design and not just pointed out as a limitation of the study. In all the text this precision must be made since the term “cohort” implies a longitudinal study and analysis, which was not done.
- RESPONSE: Thank you for your suggestions. We revised all the term about cohort in the manuscript and we also clarified our study in a cross-sectional analysis, not a cross-sectional cohort. Please find below about our changes.
- (Line 28 to 30) “We aimed to explore the associations among menopause, postmenopausal hormone therapy, and metabolic syndrome in a large community-based cohort group of Asian women.”
- (Line 30 to 31) “This is a cross-sectional study in which we enrolled women aged 30 to 70 years with sufficient information about menopausal status from the Taiwan Biobank.”
- (Line 81 to 84) “Given that community-based statistics are imperative for effective prevention and management of metabolic syndrome in postmenopausal women, the aim of this study was to investigate the associations between menopause, postmenopausal hormone therapy, and metabolic syndrome in a sizable population of Asian women.”
- (Line 88 t0 90) “This is a cross-sectional study, and data for the present study were obtained from the Taiwan Biobank (TWB), an extensive community-based research database that has been collecting information from cancer-free volunteers aged 30 to 70 years since 2008.”
- (Line 171) “The overall prevalence of metabolic syndrome in the studied population was 24.7%, including 14% in the premenopausal group and 30% in the postmenopausal group.”
- (Line 216 to 217) “Our study provides a population-based analysis of a large group of perimenopausal Asian women (N = 17,460).”
- Also, it is necessary to specify how was obtained the information about the sociodemographic and health status variables in methods section. In the study population, it should be pointed out until the year the data were obtained.
- RESPONSE: We have included a new sub-section titled "Variables" in the methods section. Please find the changes outlined below: “To gather the variables analyzed, we utilized diverse data sources, including questionnaires, physical examinations, and blood tests. The questionnaires encompassed a wide range of factors, such as age, gender, smoking, drinking, coffee consumption habits, exercise routines, educational status, marital status, and medical history. Physical examinations were conducted to assess waist circumference, body mass index, and blood pressure. Blood tests provided data on metabolic profiles, including fasting glucose, cholesterol-related profiles, and triglycerides. The data were collected from participants at the time of their enrollment in the study, which spanned from December 2008 to February 2020.” (Line 102 to 111)
- Line 159: If it is a cross-sectional, the word “cohort” generates confusion.
- RESPONSE: We revised the sentence as “The overall prevalence of metabolic syndrome in the studied population was 24.7%, including 14% in the premenopausal group and 30% in the postmenopausal group.” (Line 171) We used “population” to substitute “cohort”. The term "population" refers to the group of individuals being studied or analyzed for the prevalence of metabolic syndrome.
- Line 204. The analysis is not a cohort, it is a cross-sectional analysis of a cohort.
- RESPONSE: We revised this sentence as “Our study provides a population-based analysis of a large group of perimenopausal Asian women (N = 17,460).” (Line 216 to 217)
Reviewer 2 Report
The authors explored the associations among menopause, postmenopausal hormone therapy, and metabolic syndrome in a large community-based cohort of Asian women. Data for the study were obtained from the Taiwan Biobank, an extensive community-based research database that has been collecting information from cancer-free volunteers aged 30 to 70 years since 2008, and has recruited individuals from 29 centers throughout Taiwan. They found that menopause was associated with an increased risk of metabolic syndrome, especially for those with surgical menopause. Furthermore, postmenopausal hormone therapy was found to be associated with a lower prevalence of metabolic syndrome in the women with natural menopause. Although there are some of interests in this article, I have several major concerns about it.
1. Because it has already been reported that surgical menopause increases the risk of metabolic syndrome more than spontaneous menopause or premenopause, this study content is not novel. However, it is very commendable that this is a very large study, which has never been done before on Asian subjects.
2. We would like to see a discussion of why hormone therapy after natural menopause reduces metabolic syndrome compared to hormone therapy after surgical menopause.
3. As mentioned in the Limitation section, it would be better if more detailed information on hormone therapy, such as duration of administration, method of administration, and formulations used, could be provided.
Author Response
- We would like to see a discussion of why hormone therapy after natural menopause reduces metabolic syndrome compared to hormone therapy after surgical menopause.
- RESPONSE: No problem. We added a paragraph to mention this. Please find below “Another noteworthy finding from our study is that postmenopausal hormone therapy did not reduce metabolic syndrome in women with surgical menopause compared to natural menopause. It is well-established that surgical menopause is characterized by a heightened prevalence of metabolic syndrome, obesity, and lipid abnormalities in contrast to natural menopause (63,64). The underlying mechanisms for this disparity involve abrupt hormonal changes, a persistent low-grade inflammatory state following surgery, and modifications in adipose tissue distribution and lipid metabolism resulting from the surgical procedure (63). However, no research has specifically focused on the effects of postmenopausal hormone therapy in this population, and our study was the first to explore this topic. We proposed three reasons to explain these findings. Firstly, the timing of postmenopausal hormone therapy initiation differs between natural and surgical menopause. Natural menopause allows for a shorter gap between hormone decline and postmenopausal hormone therapy initiation, preserving metabolic function (65). Surgical menopause involves an abrupt loss of ovarian function, potentially leading to a longer gap and a less favorable metabolic response (65). Secondly, underlying health conditions, such as cancer or severe gynecological disorders, in surgical menopause may increase the risk of metabolic syndrome (66). Hormone therapy may not fully counteract the metabolic changes associated with underlying health conditions in surgical menopause. Thirdly, hormone levels decline gradually in natural menopause, allowing the body to adapt, while surgical menopause causes an immediate loss of hormones (67). Further studies are needed to validate these findings and provide a more comprehensive understanding of the effects of postmenopausal hormone therapy in women with surgical menopause.” (Line 307 to 328)
- As mentioned in the Limitation section, it would be better if more detailed information on hormone therapy, such as duration of administration, method of administration, and formulations used, could be provided.
- RESPONSE: No problem. As mentioned below: “The frequency and intensity of physical activity were not reported, and information regarding the timing, methods, and frequency of hormonal therapy use was also not collected. According to the guidelines provided by the National Institute for Health and Care Excellence (NICE) on Menopause Diagnosis and Management, hormone therapy should be prescribed at the lowest effective dose and for the shortest duration necessary to alleviate menopausal symptoms (67). The specific duration of administration can vary depending on individual needs and the symptoms being addressed. Various methods are available for hormone therapy administration, including oral, transdermal, and vaginal routes (67). The choice of method depends on factors such as the individual's symptoms, preferences, and medical history (67). In cases where a woman has undergone a hysterectomy, estrogen alone can be prescribed. However, for women with an intact uterus, a combination of estrogen and progestogen (progesterone or a progestin) is usually required to protect the lining of the uterus (67). It is important for future research to consider these details when studying hormone therapy and its effects on the prevention of metabolic syndrome.” (Line 344 to 358)
Reviewer 3 Report
Dear Authors,
I have really appreciated your manuscript because of the importance of the topic especially in this era where women duration of life is increased and an acceptable QoL is desirable.
-
Author Response
- I have really appreciated your manuscript because of the importance of the topic especially in this era where women duration of life is increased and an acceptable QoL is desirable.
RESPONSE: Thank you, I really appreciate it.
Reviewer 4 Report
The manuscript deals with the analysis of a large premenopausal and menopausal cohort of subjects. Metabolic syndrome, menopausal issues and hormone replacement therapy were evaluated if affecting the incidence of metabolic syndrome.
The manuscript is of interest
Minor criticisms:
Discussion - Table 5 is not relevant and should be omitted in the discussion. Tables and figures have to be located in the results section but this Table just give reference to previous studies that have been cited in the discussion.
Discussion - The authors should try to speculate on what mechanisms might be at the basis of the higher occurrence of metabolic sybndrome in case of surgical menopausa. Might this be dependant from a different androgen setting due to ovariectomy ?
Though not being English mother-language, the english is accettable
Author Response
- Discussion - Table 5 is not relevant and should be omitted in the discussion. Tables and figures have to be located in the results section but this Table just give reference to previous studies that have been cited in the discussion.
- RESPONSE: We completely agree, and we have deleted the table, incorporating the information into the manuscript.
- Discussion - The authors should try to speculate on what mechanisms might be at the basis of the higher occurrence of metabolic sybndrome in case of surgical menopausa. Might this be dependant from a different androgen setting due to ovariectomy ?
- RESPONSE: No problem. As mentioned below: “Another noteworthy finding from our study is that postmenopausal hormone therapy did not reduce metabolic syndrome in women with surgical menopause compared to natural menopause. It is well-established that surgical menopause is characterized by a heightened prevalence of metabolic syndrome, obesity, and lipid abnormalities in contrast to natural menopause (63,64). The underlying mechanisms for this disparity involve abrupt hormonal changes, a persistent low-grade inflammatory state following surgery, and modifications in adipose tissue distribution and lipid metabolism resulting from the surgical procedure (63). However, no research has specifically focused on the effects of postmenopausal hormone therapy in this population, and our study was the first to explore this topic. We proposed three reasons to explain these findings. Firstly, the timing of postmenopausal hormone therapy initiation differs between natural and surgical menopause. Natural menopause allows for a shorter gap between hormone decline and postmenopausal hormone therapy initiation, preserving metabolic function (65). Surgical menopause involves an abrupt loss of ovarian function, potentially leading to a longer gap and a less favorable metabolic response (65). Secondly, underlying health conditions, such as cancer or severe gynecological disorders, in surgical menopause may increase the risk of metabolic syndrome (66). Hormone therapy may not fully counteract the metabolic changes associated with underlying health conditions in surgical menopause. Thirdly, hormone levels decline gradually in natural menopause, allowing the body to adapt, while surgical menopause causes an immediate loss of hormones (67). Further studies are needed to validate these findings and provide a more comprehensive understanding of the effects of postmenopausal hormone therapy in women with surgical menopause.” (Line 307 to 328)